# Mean state and day-to-day variability of tropospheric circulation in planetary-scale barotropic Rossby waves during Eurasian heat extremes in CMIP5 models

Iana Strigunova[1,2], Frank Lunkeit[1], Nedjeljka Žagar[1], and Damjan Jelić[3]

[1]Meteorological Institute, Center for Earth System Research and Sustainability (CEN), Universität of Hamburg, Grindelberg 5, 20144 Hamburg, Germany

[2]Now at Department of Earth Sciences, Uppsala University, Uppsala, Sweden

[3]Department of Physics, Faculty of Mathematics and Physics, University of Ljubljana, Ljubljana, Slovenia; now at Department of Geophysics, Faculty of Science, University of Zagreb, Zagreb, Croatia

**Correspondence:** Iana Strigunova (iana.strigunova@geo.uu.se)

**Abstract.** Surface Eurasian heat waves (EHWs) in reanalysis datasets exhibit distinct signatures in the planetary Rossby wave circulation during extended boreal summer, particularly evident in the day-to-day variability. The representation of these signatures continues to be a challenge for climate models, despite significant advancements. This study demonstrates uncertainties in the simulated EHW-related variability in planetary-scale Rossby waves for the present-day climate and the future scenario RCP4.5 in a subset of CMIP5 models. The historical simulations represent surface EHW and the associated mean pattern of Rossby waves reasonably well, in particular the uncoupled simulations. However, the EHW signatures in day-to-day tropospheric circulation variability are not adequately reproduced. For the RCP4.5 scenario and future EHWs defined with respect to the future mean climate, models do not suggest an increase in EHWs. The associated Rossby wave circulation is considerably uncertain, with a particular lack of consistent representation of day-to-day variability. This further limits confidence in future projections of changes in EHWs. Our results suggest that intrinsic variability should be an additional component of the metrics evaluating the simulation of EHWs and their related circulation.

## 1 Introduction

Record-breaking Eurasian heat waves (EHWs) in recent years have led to devastating socioeconomic and ecological impacts (e.g. Hunt et al., 2021; Jeong et al., 2025). In the future, EHWs are expected to increase in duration, magnitude, and frequency (e.g. Seneviratne et al., 2021; De Luca and Donat, 2023) with respect to present-day climate, as a consequence of the projected global mean temperature increase due to rising greenhouse gas concentrations (e.g. Van Loon and Thompson, 2023), commonly referred to as thermodynamic driver. The future changes in atmospheric circulation associated with EHWs are much less certain (e.g. Shepherd, 2014; Barriopedro et al., 2023) due to biases in the representation of the mean state and complex multi-scale interactions in the general circulation models (GCMs) used for future projections. For example, even with a perfect atmospheric component of the coupled climate model, simulated large-scale circulation is characterised by large biases due to regional inaccuracies in simulated sea-surface temperature causing atmospheric bias teleconnections (e.g. Wang et al., 2014;

Žagar et al., 2020; Zhao et al., 2023). A multitude of regional factors like soil moisture, aerosols, vegetation, and anthropogenic influences is likely relevant for the onset and evolution of heat waves (HWs) (e.g. Barriopedro et al., 2023; Domeisen et al., 2023).

For the historical simulations, the Coupled Model Intercomparison Project Phase 5 (CMIP5) models were shown capable to represent surface temperature extremes, despite discrepancies among individual models and regions (e.g. Sillmann et al., 2013a). Surface temperature-based metrics also showed that improvements in spatial patterns, frequency, intensity and duration of heat extremes from CMIP5 to CMIP6 are limited in comparison with the observational datasets (Thorarinsdottir et al., 2020; Wehner et al., 2020), although CMIP6 median was found more skilful than CMIP5's (Fan et al., 2020; Kim et al., 2020;

Hirsch et al., 2021). However, the relation between surface heat extremes and anomalies in the atmospheric circulation remains uncertain, despite improvements in more recent CMIP phases (e.g. Vautard et al., 2023; Lembo et al., 2024). The variety of variables and thresholds used to define surface heat extremes might partially explain differences in statistics of HW-associated circulation in the CMIP models in various studies.

    Uncertainties persist about future changes in tropospheric circulation associated with surface heat extremes, which is the

35 subject of this paper. For example, the latest IPCC report (IPCC2021, chapter 8) places medium confidence in the increase of amplitudes of stationary waves, which are found to be connected with hot extremes over Eurasia and in the Northern Hemisphere in general (e.g. Screen and Simmonds, 2014; Yuan et al., 2017). The link between atmospheric blocking and HWs seems to be well represented in the CMIP5 large-ensemble (e.g. Schaller et al., 2018; Brunner et al., 2018; Jeong et al., 2022), which has been shown for the 2018 EHW (Li et al., 2020) and in the regional study over China (Wang et al., 2019). Given that

the link is realistic, it is important to better understand uncertainties in projected circulation changes in the CMIP models in relation to trends in the global mean surface temperature (Lee et al., 2021).

    The present paper contributes to this question by investigating circulation variability in planetary-scale Rossby waves associated with EHWs in historical simulations and a future scenario simulated by a subset of CMIP5 models. An important factor affecting simulated tropospheric variability is the model bias. For example, Luo et al. (2022) demonstrated that biases in the

45 upper-tropospheric circulation significantly affect surface fields in the models. The authors concluded that climate models are useful in studying present and future Rossby waves, but associated extremes on the surface should be diagnosed with caution.

    Existing uncertainties may be partially explained by the large number of metrics used to identify surface heat extremes, reflecting the complex interactions of underlying physical mechanisms (e.g. Horton et al., 2016; De Luca and Donat, 2023) and the varied needs of different scientific communities (Naomi et al., 2024). These diverse HW definitions often combine

temperature with other meteorological variables like relative humidity: for instance, the use of wet-bulb temperature to compute the US Weather Service Heat Index from Buzan et al. (2015). Metrics vary based on absolute or relative thresholds and whether characteristics like duration, intensity, frequency, and spatial extent are considered (such as the Heat Wave Intensity Duration Frequency Curve from Mazdiyasni et al. (2019) or the Heat Wave Magnitude Index from Russo et al. (2014)). The specific goal of a study also influences the metric; for example, cumulative heat for health impacts (Perkins-Kirkpatrick and Lewis, 2020)

or the timing of the HW season for ecosystem impacts (Sippel et al., 2016). To unify them in one framework, sets of indices

are proposed by the Expert Team on Climate Change Detection (https://www.wcrp-climate.org/etccdi) and the Expert Team on Climate Information for Decision-making (https://climpact-sci.org/indices/).

Furthermore, to assess future changes, some of these metrics are typically based on parameters estimated from present-day conditions (e.g. Sillmann et al., 2013b). In this case, changes in the mean climate may affect the scores for the surface extremes diagnosed by the metrics (e.g. Perkins, 2015). For example, a mean warming may lead to an increase in the number of EHWs although the variability stays the same. Since the interaction between surface and atmospheric circulation may dominantly occur on the time scale of the event, this makes the analysis of the surface-atmosphere link and its potential change challenging.

The present study is a follow-on of Strigunova et al. (2022, hereafter Setal2022) who analysed Rossby wave submonthly variance in four modern reanalysis datasets during EHWs. Setal2022 showed that a reduction of intramonthly Rossby wave variance at the zonal wavenumber $k = 3$ during EHWs is consistent with persistent large-scale circulation anomalies associated with blocking. The reduction of the variance coincides with an increased skewness of the variability of the Rossby wave mechanical energy at planetary scales ($k = 1 - 3$). Due to the barotropic structure of the boreal summer troposphere during EHWs, we focus on the troposphere-barotropic Rossby waves. We explore the following two questions:

- To what extent do the CMIP5 models represent the statistics of the tropospheric barotropic planetary-scale Rossby waves during EHWs?

- What are the projected changes in the variability of tropospheric barotropic planetary-scale Rossby waves during EHWs, given a high confidence in a surface temperature increase?

Addressing these questions requires the three-dimensional structure of Rossby waves in terms of wavenumbers. This is obtained following the methodology of Setal2022 which projects the global circulation projection onto the complete set of orthogonal Rossby and inertia-gravity modes using the normal-mode function approach (e.g. Kasahara and Puri, 1981; Tanaka and Kung, 1988; Žagar et al., 2015) and retains planetary-scale Rossby modes with the troposphere-barotropic structure. We analyse events associated with the EHWs as defined in terms of the Eurasian near-surface temperature (2-meter temperature; T2m). As stated above, defining EHWs relative to the respective mean climate is necessary when focusing on the link between EHWs and the Rossby wave circulation on the time scale of the EHW events. Therefore we use metrics that are not directly influenced by warming.

A subset of CMIP5 models is used, as available from the archive of three-dimensional (3D) circulation projection by the MODES software (Žagar et al., 2015). Given a marginal difference in the blocking frequency in CMIP5 and CMIP6 models (Doblas-Reyes et al., 2021) and uncertainties in climate projections of atmospheric blocking patterns involved in heat wave formation (Gulev et al., 2021), the dataset suffices for the first study aiming at global 3D wave-space diagnostic of HWs in CMIP models. Uncoupled atmospheric simulations forced by the observed sea-surface temperature (SST) and historical coupled simulations are first compared with the reanalysis data. Then, we compare historical simulations with the Representative Concentration Pathways (RCPs) scenario RCP4.5 which is considered a moderate and plausible scenario of future climate (Thomson et al., 2011; Moss et al., 2010; van Vuuren et al., 2011).

Further details of the datasets and the methodology are presented in Section 2. The climatology of the T2m that defines
EHWs in the models' historical simulations and the RCP4.5 scenario is presented in Section 3. Section 4 evaluates circulation
anomalies associated with EHWs. Their structure and the characteristics of their day-to-day variability in historical simulations
are compared with reanalysis data, and changes observed in the RCP4.5 scenarios are discussed. Section 5 contains conclusions.

## 2    Data and Methods

In this Section we describe the data we used, which consists of CMIP5 simulations and reanalyses (Section 2.1), as well as the
methods used to determine the surface EHWs and the Rossby wave circulation associated with them (Section 2.2).

### 2.1    Data

Focusing on EHWs, we analyse the extended boreal summer season from May to September (MJJAS). We use a subset of
CMIP5 models that had outputs available on model levels to apply wave decomposition on terrain-following levels (see Section
2.3 below). No further selection criteria are applied. Our model subset consists of the CNRM-CM5 (Voldoire et al., 2013),
the GFDL-CM3 (Donner et al., 2011), the MIROC5 (Watanabe et al., 2010) and the MPI-ESM-LR (Giorgetta et al., 2013).
Although our selection is based only on the availability of the data, we note that the four models are among the six models
identified by Basharin et al. (2016) as climate models that best reproduce the historical behaviour of surface air temperature
over greater Europe, selected from the CMIP5 project using a performance-based selection method.

Given the relatively small number, our model subset reasonably represents the spectrum of the CMIP5 simulations with
regard to EHWs and atmospheric blockings. Concerning the EHWs, Hirsch et al. (2021) provide a thorough comparison of
individual CMIP5 and CMIP6 models in their supporting information. Our four models appear to lie well within the range
spanned by all CMIP5 models with respect to the bias skill scores for HW frequency, length of the longest HW, average HW
intensity and cumulative heat. The same appears to be true for the representation of Northern Hemisphere blocking events. A
comparison of the blocking frequencies of individual CMIP5 models including our four models is presented in the supporting
information of Dunn-Sigouin and Son (2013), together with a comparison of the 500-$\mathrm{hPa}$ zonal winds and variability.

Individual members of CMIP5 simulations are the historical coupled simulation (HIST), historical simulation forced by the
observed SST (Atmosphere-only model simulations; AMIP) and a future projection following the scenario RCP4.5, which is
considered a moderate and plausible scenario of future climate (Thomson et al., 2011; Moss et al., 2010; van Vuuren et al.,
2011). An overview of the CMIP5 experimental designs can be found in Taylor and Meehl (2012).

Our CMIP5 datasets involve daily data for the historical climate represented by the 26-year period 1980-2005 and the
projected future by the 31-year period 2070-2100. The longer period for the RCP4.5 is chosen to have a larger sample size.
However, using a 26-year period (2075-2100) as for HIST shows almost the same results (see Section 3).

We compare the CMIP5 simulations with reanalysis data. In most parts of this study (Sections 3, 4.1), we use the European
Reanalysis ERA5 (Hersbach et al., 2020). In addition, three other reanalyses, ERA-Interim (Dee et al., 2011), the Japanese 55-
year Reanalysis JRA-55 (Kobayashi et al., 2015), and the Modern-Era Retrospective analysis for Research and Applications

MERRA (Rienecker et al., 2011) are employed to assess changes in day-to-day variability during EHWs (Section 4.2) because of the small sample size for each dataset.

We mainly use a 40-year period (1980 to 2019) for ERA5 and a 35a period (1980–2014) for the other three reanalyses, where pre-processed data for the filtered troposphere barotropic Rossby-wave circulation were available (see Section 2.3). We use the entire periods (40 and 35 years, resp.) to consider the largest possible datasets, given the rarity of EHWs. However, results using the 26-year period given by the CMIP5 AMIP and HIST simulations show only minor differences.

## 2.2 The Eurasian surface heat waves

Following Setal2022, we identify EHWs using the mean daily T2m averaged over Eurasia. Averaging is done using data on the individual input grids (Table 1). Our method is similar to the methods applied by Perkins-Kirkpatrick and Gibson (2017) and Ma and Franzke (2021). EHW events are identified by three or more consecutive days of positive anomalies. The surface EHW detection is performed independently for every model run. The Eurasian region is defined between $35°N - 65°N$ and $10°W - 60°E$ and is limited by the Ural mountains. The 95th percentiles of the averaged time series are subtracted to remove the annual cycle, and only positive anomalies are considered. In order to interpret the following results, it is important to note that our method is independent of the mean state.

We assess the EHWs in Section 3 by the metrics of Perkins-Kirkpatrick and Gibson (2017). The metrics comprise the number of EHW days, the number of EHW events, the average event duration, the maximum event duration, and the peak intensity. However, details in such indices are sensitive to thresholds, in particular the maximum duration and the peak intensity. To add a less sensitive metric, we extended the set by adding the maximum temperature and the difference between the maximum temperature and the mean temperature. Accounting for the different lengths of our datasets, we normalise the total numbers of EHW days and events to a period of 10 years.

## 2.3 The planetary-scale troposphere-barotropic Rossby waves

Troposphere-barotropic Rossby waves are filtered following the methodology outlined in Setal2022. The global 3D circulation, given by the horizontal wind components $u$, $v$ and the geopotential height $h$, is projected onto a complete set of orthogonal modes using the normal-mode function approach (e.g. Kasahara and Puri, 1981; Tanaka and Kung, 1988; Žagar et al., 2015). The linear wave decomposition represents atmospheric circulation as a superposition of the zonal-mean flow and waves. In the normal-mode function framework, the decomposition is multivariate and produces a set of modes associated with two main dynamical regimes: Rossby waves (linearly balanced regime) and inertia-gravity waves (linearly unbalanced regime). Such a decomposition implemented in the MODES software (Žagar et al., 2015) produces time series of the complex expansion coefficients for the two types of motions, where each coefficient is characterised by the zonal wavenumber $k$, the meridional mode index $n$ and the vertical-mode index $m$. Index $m$ is associated with the vertical structure functions with increasing complexity for larger $m$.

For the projection, we use CMIP5 datasets in the resolutions given in Table 1 and choose a spectral truncation ($k$, $n$ and $m$) tuned to the individual input. For the projection of all reanalyses data we use data interpolated onto a regular Gaussian

grid with 256 by 128 grid points in the zonal and meridional directions, respectively, and 43 predefined $\sigma$ (pressure divided by surface pressure) levels. The spectral resolution of the output is $k$=100, $n$=49 and $m$=27.

The Rossby modes with vertical structure functions which are quasi-barotropic within the troposphere (no zero crossing) define the troposphere-barotropic Rossby waves; see Žagar et al. (2015) and Setal2022 for details. For all reanalyses and for GFDL-CM3 and MPI-ESM-LR, which have a high model top, the number of troposphere-barotropic vertical modes is five ($m = 1 - 5$), whereas for CNRM-CM5 and MIROC5, only the first two vertical modes have no zeros in the troposphere. From these modes we select those with zonal wavenumbers $k$=1 to 3 to represent the planetary scale circulation. Back transformation into grid point (physical) space provides the filtered planetary-scale troposphere-barotropic Rossby wave field, which we use to study the circulation during EHWs (Section 4.1). For this part, all datasets are regridded to the 256 by 128 grid of the reanalyses.

In addition, the square of the absolute value of the complex expansion coefficient represents the total mechanical energy of the particular mode, where the mechanical energy is the sum of kinetic and available potential energy (Žagar et al., 2015). Thanks to the 3D orthogonality of normal modes, the energies of the individual modes are additive. Energy anomalies are calculated relative to the climatology, which is defined for each calendar day of the extended boreal summer (MJJAS), and normalised by the climatological standard deviation (i.e. by variability). We use the time series of the mechanical energy to assess the day-to-day variability of troposphere-barotropic Rossby wave circulation associated with EHWs in terms of the probability density function (PDF), in particular the skewness (Section 4.2).

**Table 1.** ERA5, as an example of an observational dataset, and CMIP5 model parameters and their truncations for wave decomposition by MODES. The zonal, meridional and vertical truncations are denoted $K$, $N$ and $M$, respectively. The regular Gaussian grid is denoted by $F$ and the resolution is determined by grid spacing, which depends on the number of grid points.

| Model | AMIP/HIST/RCP4.5 | Hor. resolution (N. of pts) | No. levels | Top level (hPa) | MODES trunc. ($K \times N \times M$) |
|---|---|---|---|---|---|
| ERA5 | - | F64 ($256 \times 128$) | 43 | 0.5 | $100 \times 49 \times 27$ |
| CNRM-CM5 | yes/yes/yes | F64 ($256 \times 128$) | 28 | 10 | $64 \times 64 \times 20$ |
| GFDL-CM3 | yes/yes/yes | F36 ($144 \times 72$) | 45 | 0.01 | $72 \times 36 \times 33$ |
| MIROC5 | yes/yes/yes | F64 ($256 \times 128$) | 37 | 3.5 | $80 \times 64 \times 25$ |
| MPI-ESM-LR | yes/no/no | F48 ($192 \times 96$) | 44 | 0.01 | $60 \times 48 \times 32$ |

## 3 The statistics of Eurasian surface heat waves

The Eurasian T2m distribution for all simulations and, for comparison, ERA5 is presented by box plots in Fig. 1. For the present-day climate, the interquartile ranges (IQRs) shown in the box plots of HIST and AMIP simulations of CMIP5 overlap with the one of ERA5. All model medians of the AMIP simulations are higher than those of the HIST runs. Among the models, the coupled GFDL-CM3 has a lower median compared to ERA5, while CNRM-CM5 and MIROC5 medians are higher, with the median for MIROC5 found to be outside of the ERA5 interquartile range. Similar behaviour (large positive biases in near-

surface temperature) was observed by Flato et al. (2013). Sillmann et al. (2013a) found that the MIROC5 model performs slightly worse than the investigated CMIP5 models with respect to four reanalyses.

The position of the whiskers indicates the extent to which the maxima and minima are stretched beyond the IQR. Due to the rightward shift of the overall distributions in the uncoupled compared to the coupled runs, it is difficult to estimate the difference in the day-to-day T2m variability, except for CNRM-CM5, where the variability is smaller and closer to the range of ERA5 in uncoupled simulations. The variability can also be estimated by comparing IQRs. For example, for models with a higher median (CNRM-CM5 and MIROC5), IQRs are larger in comparison with ERA5 data as well. This does not apply to other models. For example, the GFDL-CM3 and the MPI-ESM-LR show small IQRs in all runs but contain outliers indicating large negative biases and a more skewed distribution. This skewness is a prominent feature found in ERA5 and all models. All datasets are left-skewed as the upper quartile of positive anomalies (indicated by the right whiskers) are limited while the lower quartiles of negative anomalies (left whiskers) extend to the range given by the IQR.

All RCP4.5 scenarios show a substantial increase in European T2m, with the lowest warming for the CNRM-CM5. This warming is in line with Basharin et al. (2016) who found typical changes over Europe of about 2 to 4 K by the end of the 21st century for RCP4.5 compared with 4 to 8 K for RCP8.5. However, despite the substantial warming, visual comparison between the RCP4.5 and the HIST results does not seem to indicate a large change in the variability.

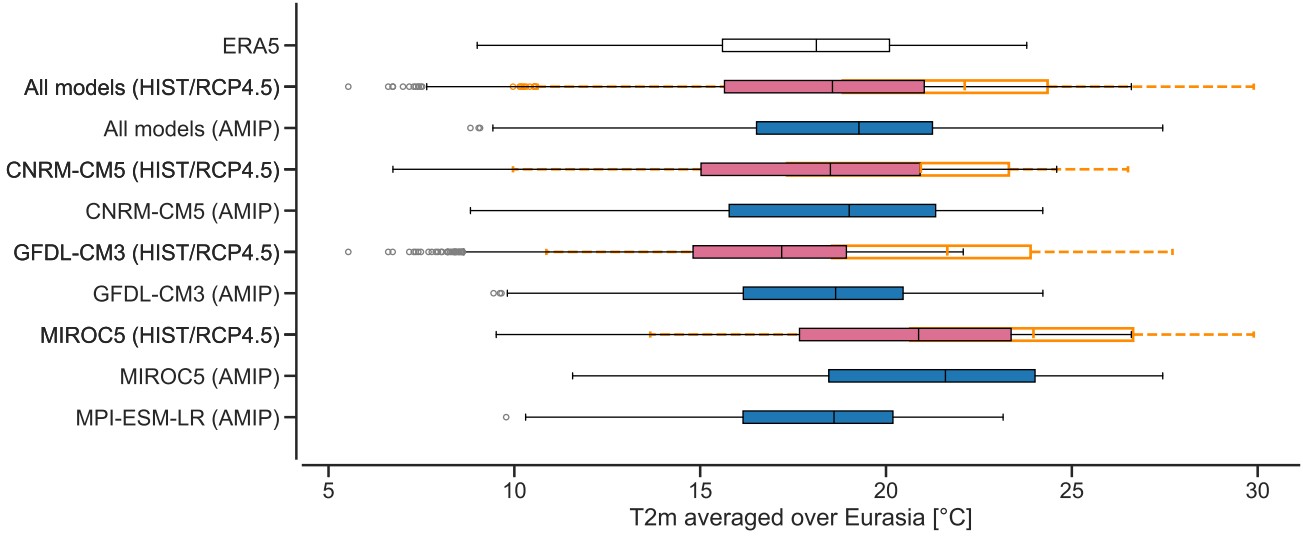

**Figure 1.** Box plots of daily near-surface air temperature (T2m) averaged over Eurasia. ERA5 is represented as a white box plot and CMIP5 models are shown as dashed and coloured boxes: coupled simulations (HIST) are red boxes, uncoupled (AMIP) are in blue and the future scenarios (RCP4.5) are in orange (dashed). Note that due to the asymmetry of the data the whiskers end at minimum and maximum values inside the maximum range (Q1-1.5IQR, Q3+1.5IQR), where the lower (Q1) and upper (Q3) quartiles frame the boxes and the interquartile range is IQR=Q3-Q1. Interquartile ranges (IQRs) are displayed as whiskers with boxes framed with vertical lines (25th (Q1) and 75th (Q3) percentiles).

The results for the EHW metrics are shown in Table 2. Overall, the models show similar results in all metrics for the present-day climate and are close to the ERA5 results for the 26-year period. However, larger differences exist for the mean duration and the maximum duration considering the 40-year period. These differences are explained by the extreme 2010 Russian heat wave. EHWs of such intensity are not present in the CMIP5 simulations nor in the ERA5 data for the 26-year period. There are no systematic differences between coupled (HIST) and uncoupled (AMIP) simulations, except for CNRM-CM5, where all parameters are smaller in the AMIP simulations. We conclude that frequency and duration of EHWs are reasonably represented by the T2m in the considered CMIP5 models, however with differences in the max temperature up to 3 and 4 °C in the MIROC5 model. Using different metrics, Hirsch et al. (2021) found reasonable agreement for the duration, an under-prediction of frequency and an overestimate of the magnitude by most CMIP5 and CMIP6 models with only minor differences between CMIP5 and CMIP6. For RCP4.5, CNRM-CM5 shows a slight decrease in the number and in the maximum duration of EHWs, while the other two models show no change in number and a substantially longer maximum duration. Only small differences exist between the 30-year and the 26-year period for RCP4.5 (R1 and R2). Again, we note that, in our analysis, a change in the mean climate does not directly affect the metrics.

**Table 2.** Heat wave metrics. EHW days and events are the total number normalised to the total number in a 10-year interval in the period 1980 to 2005 for AMIP and HIST, 2070 (2075) to 2100 for RCP4.5, and ERA5 for 1980 to 2019 (2005), which serves as a reference dataset. $T_{max}$ is the maximum temperature observed in an EHW and $\Delta T = T_{max} - T_{mean}$. The values for the CMIP5 models are for AMIP, HIST and RCP4.5 two periods (A/H/R1 [R2]), respectively. For ERA5, the values in brackets refer to the 26-year period. For MPI-ESM-LR only the AMIP simulation is analysed.

| Reanalysis/Model | HW days | HW events | mean duration (days) | max duration (days) | $T_{max}$ (°C) | $\Delta T$ (°C) |
|---|---|---|---|---|---|---|
| ERA5 | 52 [41] | 7 [9] | 7.6 [4.6] | 26 [12] | 23.8 [23.0] | 6.1 [5.6] |
| CNRM-CM5 (A/H/R1[R2]) | 39/51/40 [50] | 8/10/7 [10] | 4.6/4.9/5.4 [5.2] | 12/14/12 [12] | 24.2/24.6/26.5 [26.5] | 5.8/6.9/6.4 [6.4 |
| GFDL-CM3 (A/H/R1[R2]) | 42/43/38 [47] | 7/7/6 [7] | 5.7/5.8/6.2 [6.5] | 15/12/30 [30] | 24.2/22.1/27.2 [27.7] | 6.1/5.4/6.7 [6.7 |
| MIROC5 (A/H/R1[R2]) | 52/51/41 [48] | 10/9/8 [8] | 5.2/5.5/5.3 [5.7] | 11/13/17 [17] | 27.4/26.6/29.9 [29.9] | 6.3/6.3/6.5 [6.5 |
| MPI-ESM-LR (A) | 53 | 8 | 6.6 | 15 | 23.2 | 5.2 |

In summary, we find minor differences in the T2m distributions for the historical (AMIP and HIST) simulations in comparison with ERA5, except for large deviations detected in one model (MIROC5). Most importantly, the IQRs are similar and all distributions are skewed, linked to the similarity in positive T2m anomalies relative to the respective medians (Fig. 1). Overall, there is a shift in T2m for RCP4.5 for all models to the extent that the median is on the same level as Q3 for the present climate, in line with IPCC AR6 (IPCC, 2023). EHWs are represented by all models in a similar way and in good agreement with ERA5. Almost no changes are found for the number of EHWs in the future scenario, but a longer duration is simulated by two models.

## 4 Tropospheric circulation anomalies during EHWs

Now we ask how the planetary signals of barotropic Rossby waves associated with EHWs are represented by the subset of CMIP5 models. First, we present the Rossby wave spatial structures in historical runs and the RCP4.5 projection. This is followed by PDFs of the day-to-day variability during EHWs in comparison with climatology, following the methodology from Setal2022.

### 4.1 The spatial structure of planetary-scale Rossby waves

We investigate the spatial structure of barotropic Rossby waves associated with surface EHWs discussed in Section 3. We present the 500 hPa level geopotential and horizontal wind climatology and composites during EHWs in the models and compare them among the three runs, against their MJJAS climatology and against ERA5. The climatological averages for HIST and AMIP for all models are compared with ERA5 in Fig. 2 (for the differences between the CMIP5 simulations and ERA5 see Fig. S1 in the supplement). While relatively small differences between HIST and AMIP are present for all models (Fig. S2 in the supplement), larger differences can be found among the models and between the models and ERA5. Although in reasonable agreement for the Atlantic-European sector, all four models overestimate the strength of the stationary wave pattern compared to ERA5. This leads to a more dominant wavenumber two structure with a pronounced high over western America. This bias is slightly larger for HIST than for AMIP and most distinct for the MIROC5 simulations. For a more quantitative comparison, Table 3 provides the Root Mean Square Errors (RMSEs) and Anomaly Correlations (ACCs) for northern hemispheric 500-hPa climatologies of the CMIP5 HIST and AMIP simulations with respect to ERA5. The measures support our assessment by showing that AMIP simulations exhibit, with very few exceptions, higher values for ACC and lower values for RMSE compared to HIST.

Except for CNRM-CM5 HIST, all simulations show a strengthening of the climatological European height during EHWs and a shift of the Pacific low towards eastern Asia, both in accordance to ERA5 (Fig. 3; for differences see Fig. S3 in the supplement). This reflects a typical European blocking situation. Both changes are larger and in better agreement with ERA5 for AMIP than for HIST. In contrast to CNRM-CM5 AMIP, almost no changes during EHWs are found over Europe for CNRM-CM5 HIST. While in GFDL-CM3 HIST and both MIROC5 simulations the wavenumber two structure of the climatological average persists during EHWs, the high over North America is diminished in GFDL-CM3 AMIP, which is in better agreement with ERA5. As for the climatology, the RMSEs and ACCs support our qualitative assessment, with better agreement with ERA5 for the AMIP simulations compared to HIST (Table 3, values in parentheses).

Overall AMIP outperforms HIST, and MIROC5 has the largest deficits in the simulated patterns. While hard to identify causes of differences between the AMIP and HIST run, they may be related to a lack of energy dissipation from the atmosphere to the ocean or bias teleconnections in the model associated with errors in simulated SST in remote regions (e.g. Zhao et al., 2023, and references therein).

For RCP4.5, the averaged planetary Rossby wave circulation shows a reduction of the European height and the Pacific low together with a deepening of the Icelandic low for all models, but to varying degrees (Fig. 4a-c; for differences see Fig. S4 in

the supplement). The circulation changes during EHWs vary among the three models and differ from those in HIST (Fig. 4d-f,
for differences see Fig. S5 in the supplement). CNRM-CM5 and MIROC5 show a strengthening of the European height and a
westward shift of the Pacific low. Instead, the circulation in GFDL-CM3 appears mostly unaffected during EHWs.

   In summary, uncoupled simulations (AMIP) systematically outperform coupled (HIST) simulations in reproducing the pat-
terns and the magnitudes of the anomalies observed during present-day EHWs. For the future (RCP4.5), the simulated EHW
anomalies substantially differ from the HIST simulations. The greatest qualitative agreement is found for MIROC, which, on
the other hand, has the largest deficits in simulating the present-day climate.

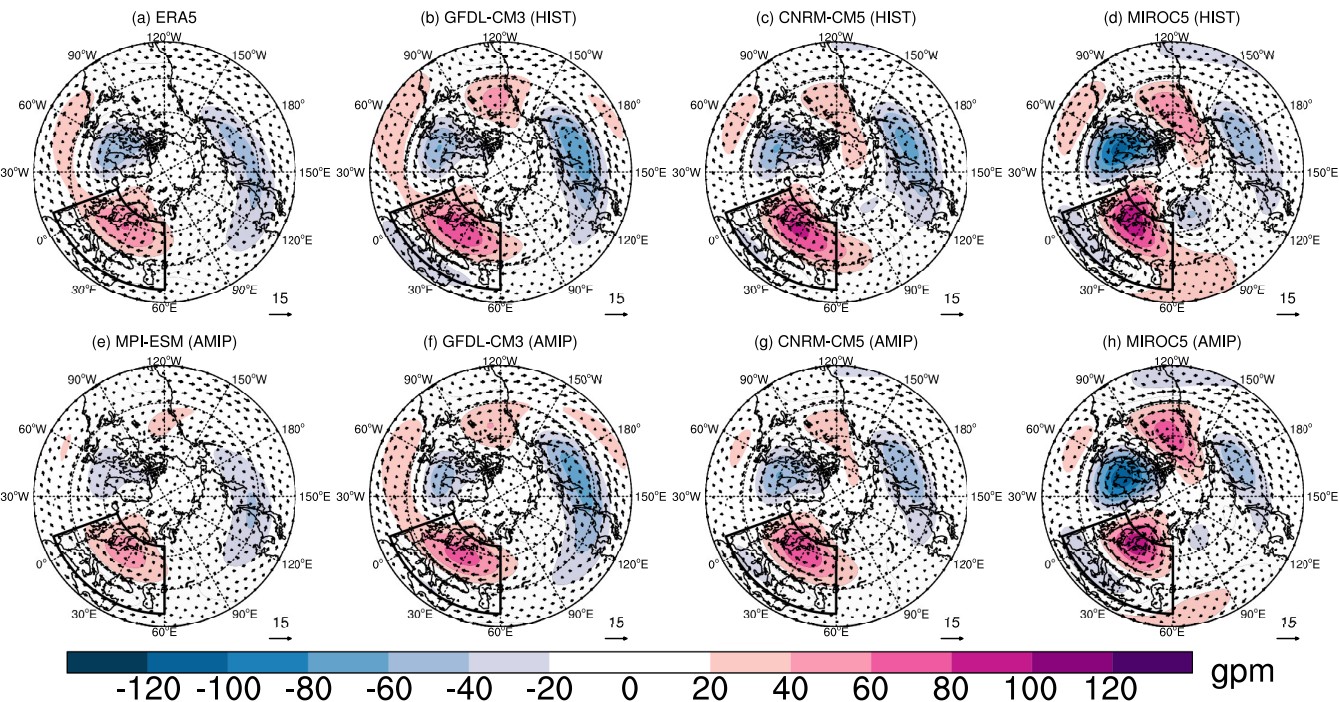

**Figure 2.** Climatologies of MJJAS mid-troposphere (500 hPa) planetary Rossby-wave circulation (geopotential height anomalies and winds)
for ERA5 (a), the HIST simulation of GFDL-CM3 (b), CNRM-CM5 (c) and MIROC5 (d), and the AMIP simulation of MPI-ESM (e), GFDL-
CM3 (f), CNRM-CM5 (g) and MIROC5 (h). The climatologies are computed for the period 1980 to 2005 for the simulations and 1980 to
2019 for ERA5. Geopotential height anomalies are shaded, wind speed in m/s is indicated by the arrow length.

## 4.2   Day-to-day variability

Now we are looking into day-to-day variability during the EHWs compared to climatology, defined by the PDF of the nor-
malised mechanical energy (see Section 2.2). Setal2022 noted an increase in the probability on the right-side tail (higher than
normal) and a shift of the maximum towards the left (lower than normal) of the energy anomaly distribution for the plane-
tary scales during EHWs in reanalyses which was associated with an increase in the skewness. This points to a reduction in

**Table 3.** Anomaly Correlation (ACC) and Root Mean Squared Error (RMSE) for the CMIP5 HIST and AMIP simulations with respect to ERA5. RMSE and ACC are provided for the northern hemispheric 500-hPa climatologies, CLIM (shown in Fig. 2) and, in parentheses, for the respective difference between EHW composites and climatologies, DIFF (shown in Fig. S3 in the supplement).

| ACC/RMSE CLIM (DIFF) | MPI-ESM-LR AMIP | CNRM-CM5 HIST | CNRM-CM5 AMIP |
|---|---|---|---|
| h'@500hpa | 0.93 (0.79) / 6.2 (8.13) | 0.89 (0.12) / 8.98 (13.25) | 0.91 (0.62) / 7.18 (11.35) |
| u'@500hpa | 0.9 (0.72) / 0.79 (0.82) | 0.87 (0.36) / 0.99 (0.97) | 0.83 (0.41) / 1.12 (1.27) |
| v'@500hpa | 0.93 (0.79) / 0.29 (0.44) | 0.93 (0.11) / 0.4 (0.78) | 0.94 (0.72) / 0.31 (0.55) |
| GFDL-CM3 HIST | GFDL-CM3 AMIP | MIROC5 HIST | MIROC5 AMIP |
| 0.89 (0.45) / 9.2 (11.38) | 0.91 (0.71) / 7.6 (10.4) | 0.82 (0.24) / 14.1 (13.66) | 0.82 (0.68) / 13.94 (10.05) |
| 0.85 (0.27) / 1.3 (1.08) | 0.92 (0.58) / 0.96 (1.12) | 0.83 (0.07) / 1.38 (1.3) | 0.78 (0.36) / 1.53 (1.19) |
| 0.9 (0.44) / 0.43 (0.64) | 0.92 (0.74) / 0.33 (0.55) | 0.9 (0.5) / 0.78 (0.65) | 0.92 (0.76) / 0.77 (0.48) |

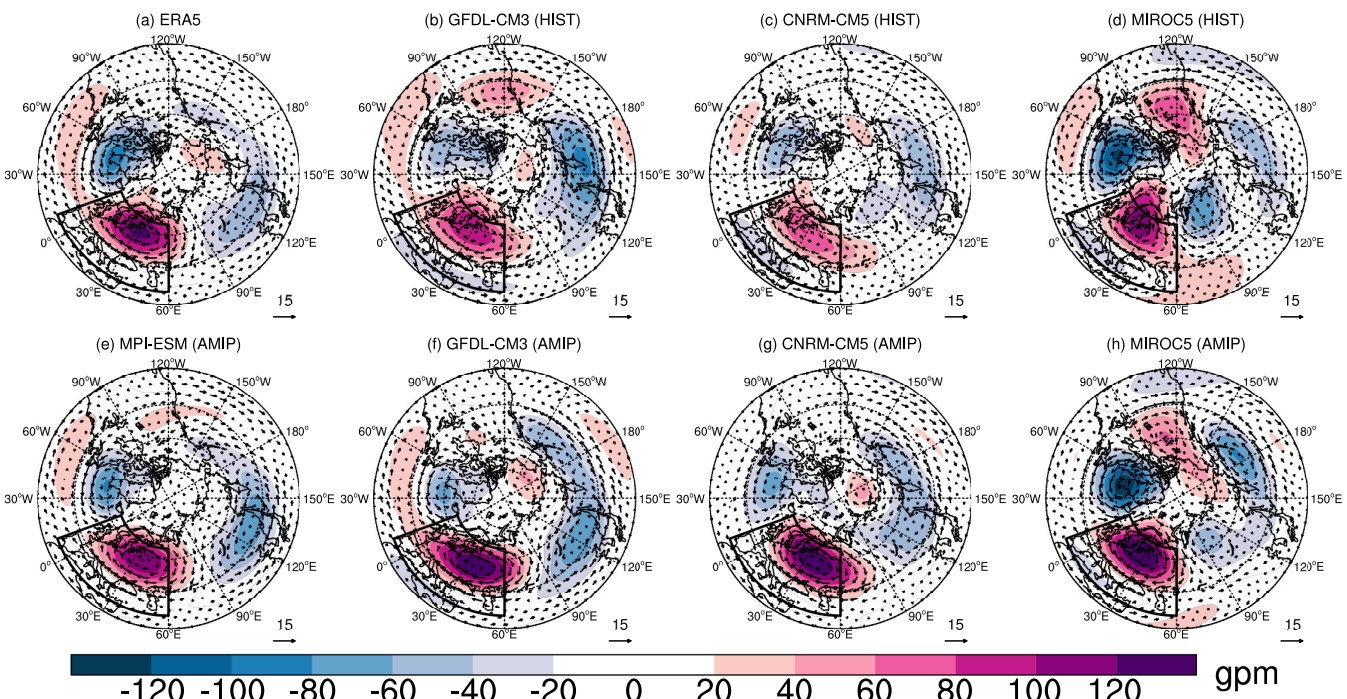

**Figure 3.** EHW composites of MJJAS mid-troposphere (500 hPa) planetary Rossby-wave circulation (geopotential height anomalies and winds) for ERA5 (a), the HIST simulation of GFDL-CM3 (b), CNRM-CM5 (c) and MIROC5 (d), and the AMIP simulation of MPI-ESM (e), GFDL-CM3 (f), CNRM-CM5 (g) and MIROC5 (h). The numbers of EHW events are given in Table 2 except for ERA5 where we use 28 EHW events in the period 1980 to 2019. Geopotential height anomalies are shaded. Wind speed in m/s is indicated by the arrow length.

variability which was confirmed with submonthly variance spectra. It was suggested that this change in day-to-day variability reflects changes in the internal atmospheric dynamics during EHWs.

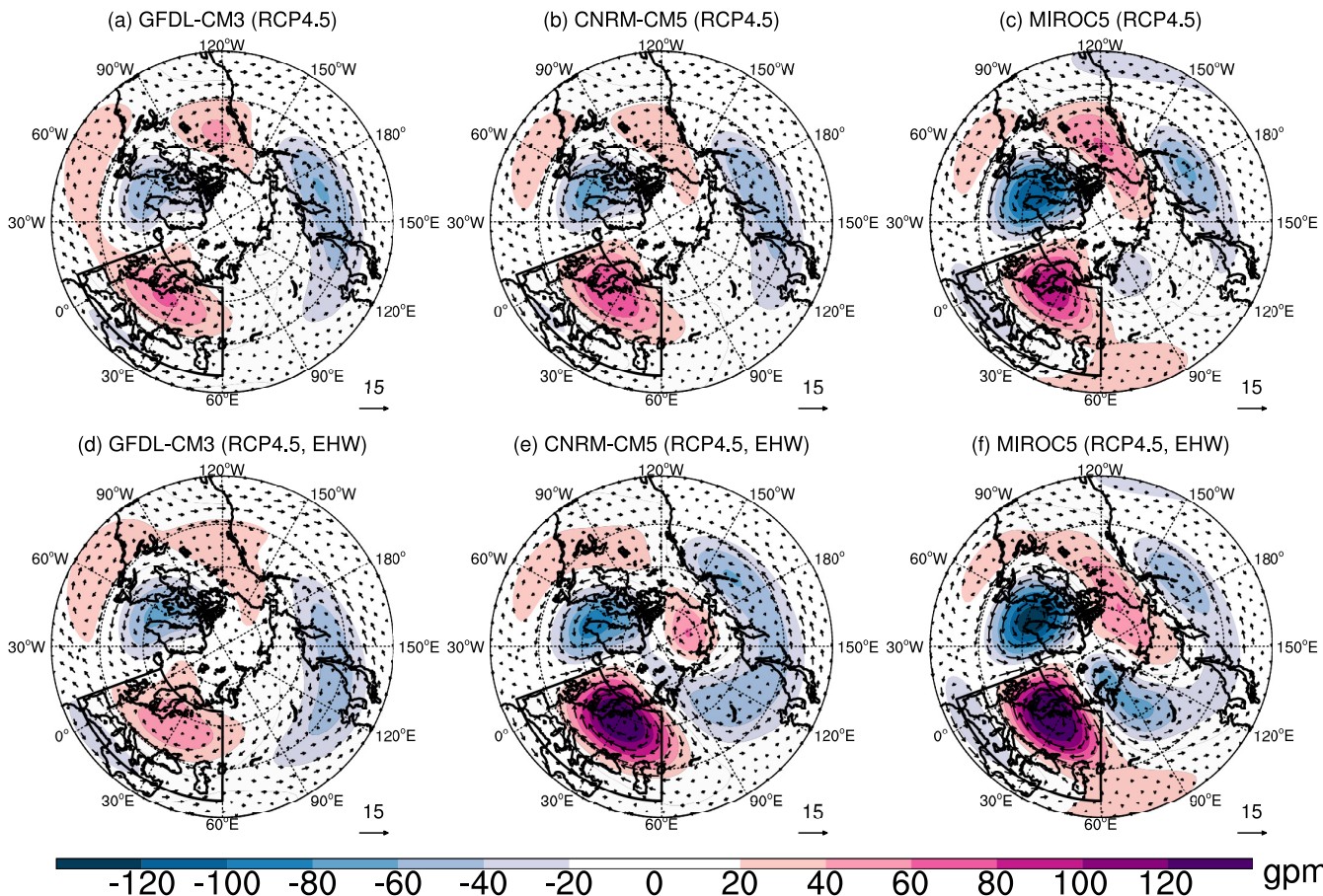

**Figure 4.** Climatology (a-c) and EHW composites (d-f) of MJJAS mid-troposphere (500 hPa) planetary Rossby-wave circulation (geopotential height anomalies and winds) for the RCP4.5 simulation of GFDL-CM3 (a,d), CNRM-CM5 (b,e) and MIROC5 (c,f). The climatologies are computed for the period 2070 to 2100. The numbers of EHW events are given in Table 2. Geopotential height anomalies are shaded. Wind speed in $\mathrm{m/s}$ is indicated by the arrow length.

The normalised energy PDFs for the reanalysis data and the model subset are shown in Fig. 5. Here, the model subset combines the data from all models. For the model subset we observe a broader and flatter distribution compared to the reanalysis. Although the differences in the skewness are not large, we note a higher skewness for all days in the HIST and AMIP runs, respectively, compared to the reanalysis, with a lower value for AMIP. The skewness of RCP4.5 for all days is lower than for both HIST and AMIP. More importantly, the substantial increase in skewness for the reanalysis is not found for the model subset. The CMIP5 models do not reproduce the increase in tails of the PDFs found in the reanalysis. The skewness only slightly increases for HIST and even shows a strong decrease for AMIP and a moderate decrease for RCP4.5. The skewness for the individual models (Fig. 6) shows a very diverse picture illustrating the lack of robustness of the results among the models.

In summary, there is little or no agreement on the change of day-to-day variability during EHWs among the models and between models and reanalyses. This may indicate substantial differences between the model's internal dynamics during EHWs, e.g. the interaction between different scales.

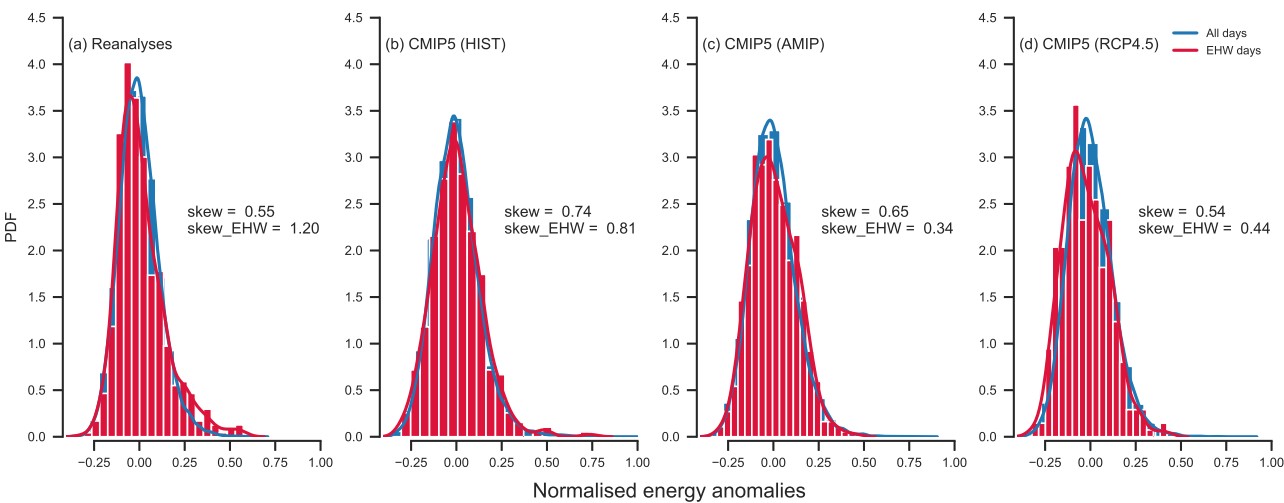

**Figure 5.** PDFs of the normalised energy anomalies on planetary scales for reanalyses (a; same as Fig. 6c in Strigunova et al. (2022)) and the CMIP5 model subset computed using all HIST (b), all AMIP (c) and all RCP4.5 (d) simulations. Blue (red) curves with shading are normalised energy anomalies for all days (only during EHWs). Note that the identification algorithm is applied for each model and simulation separately. Skewness values are indicated.

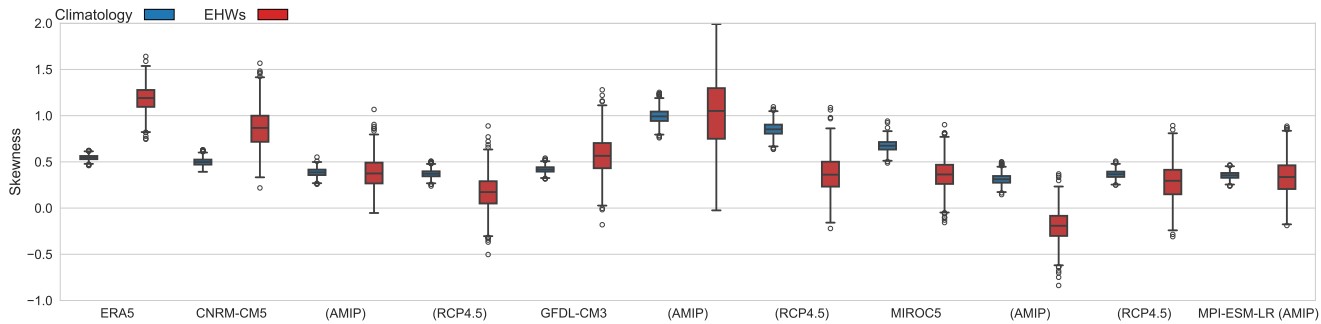

**Figure 6.** Bootstrapped skewness of the PDFs of normalised energy anomalies on planetary scales for ERA5 and each CMIP5 model with specified run (shown in parenthesis, the model name represents the HIST simulation). Blue box plots represent climatology and red box plots are during EHWs.

## 5 Summary and conclusions

We assessed troposphere-barotropic planetary Rossby waves ($k = 1 - 3$) during surface EHWs in a subset of CMIP5 models: CNRM-CM5, GFDL-CM3, MIROC5 and MPI-ESM-LR. The EHWs are defined for extended boreal summer (MJJAS) by using the near-surface air temperature (T2m). Our analysis includes both present-day conditions (coupled simulations - HIST and uncoupled simulations - AMIP), period 1980 to 2005) and a future scenario RCP4.5 for period 2070 to 2010). For the present-day climate we compared simulated CMIP day-to-day circulation variability with ERA5 reanalysis data.

Overall, we observed a reasonable agreement between the Eurasian T2m simulated by the models and ERA5, with larger deviations for one model (MIROC5). For the RCP4.5 scenario, one of the models (CNRM-CM5) showed a decrease in the number of EHWs and a decrease in their maximum duration. The other two models (GFDL-CM3 and MIROC5) showed no change in the number of EHWs but their maximum duration increased by 18 (GFDL-CM3) and 4 (MIROC5) days. Here, it should be highlighted that we define EHWs using anomalies with respect to the simulated mean climate of the respective model. That is, an increase in the mean temperature, as presented in Fig. 1, does not lead to an increase in extreme statistics, i.e. an increase in extremely warm periods at the surface, the European HWs in our case.

For the present-day climate, a relatively good agreement between the models and ERA5 was found for the planetary Rossby waves, with the exception of one model (MIROC5). However, all four models overestimated the amplitude of climatological planetary-scale circulation in MJJAS. The EHW circulation patterns in the models are also qualitatively consistent with ERA5 characteristics, with the largest differences for MIROC5. The models represent an intensification of the European high and a displacement of the Pacific low, i.e. a blocking pattern during the EHWs. This confirms the existing link between HWs and blockings documented for CMIP5 simulations (e.g. Schaller et al., 2018; Brunner et al., 2018; Jeong et al., 2022).

The uncoupled (AMIP) simulations outperform the coupled (HIST) simulations for both MJJAS climatology and, in particular, EHWs. However, little agreement was found for the day-to-day variability among the models and between the models and ERA5. Furthermore, no robust change in day-to-day variability during EHWs could be identified for the 2070-2100 period in the RCP4.5 scenario.

The results provide the following answers to the questions posed in the introduction;

– **Extent to which the CMIP5 models represent the tropospheric planetary-scale Rossby waves during EHWs:** The analysed CMIP5 model subset represents the present-day surface EHWs (T2m), as well as anomalies in the planetary-scale tropospheric Rossby waves during EHWs. However, our study found that there is little or no agreement in the change of day-to-day variability among the models and between the models and reanalysis.

– **Projected changes in the variability of planetary-scale Rossby waves during EHWs:** The models project surface warming but differ in their prediction of its statistics and associated tropospheric planetary-scale Rossby waves. In particular, very little confidence can be placed in the predicted changes in the day-to-day variability since present-day simulations already have large deficits with respect to this parameter.

Reducing prediction uncertainty requires the identification of the sources of model biases and their relative importance for the interplay between surface EHWs and atmospheric circulation including its variability. As discussed in Setal2022 for the reanalysis datasets, the changes in day-to-day variability, as identified by the skewness of the PDFs, indicate a change in internal dynamics during EHWs, for example, changes in the wave-wave or wave-mean flow interaction. Setal2022 argued that the increase in skewness for the PDFs of the planetary waves during EHWs hints to a decrease in the number of active degrees of freedom, indicating fewer independent modes involved in the circulation. Whether this can be the result of wave-wave or wave-mean flow interaction remains to be explored. Furthermore, it remains to identify possible causal relationships between the representation of the day-to-day variability in reanalysis data and the uncertainties in CMIP predictions of the surface EHW and associated planetary-scale Rossby waves and their changes in the warming climate.

In order to establish correct causality, models should provide a realistic representation of both thermodynamics and dynamics, i.e. surface EHWs and Rossby wave circulation. Previous studies have identified several possible areas for model improvements: increasing the models' resolution, the representation of orography and transient eddies, and the interaction of the atmosphere with the ocean and land (e.g. Schiemann et al., 2020; Davini and D'Andrea, 2016; Pithan et al., 2016; Martius et al., 2021). Comparing CMIP5 and CMIP6 simulations, Schiemann et al. (2020) found that higher-resolution models represent the mean blocking frequency better than low-resolution models for the Euro-Atlantic region, consistent with previous studies. Improvements can be associated with a better representation of the effects of transient eddies and the orography, partly supported by improved parameterisations of the drag (e.g. Davini and D'Andrea, 2016; Pithan et al., 2016). However, Davini and d'Andrea (2020) also argued that biases are not entirely alleviated simply by improving resolution. Scaife et al. (2011) illustrated that atmosphere-only simulations exhibit a better blocking climatology, while Davini and D'Andrea (2016) only found a weak influence of the SST. Michel et al. (2023) showed that the presence of a mesoscale eddy-permitting ocean model increases the realism of simulated blocking events. This may indicate that, beside the large-scale forcing by SST, ocean-atmosphere interactions on smaller temporal and spatial scales could play a considerable role in the representation of EHWs. Finally, atmosphere-land feedback mechanisms, in particular atmosphere-soil moisture feedbacks, have been identified as an important parameter for the evolution of EHWs (e.g. Fischer et al., 2007) and the planetary Rossby wave circulation (e.g. Martius et al., 2021).

Even though the representation of HWs in CMIP models has improved, considerable challenges remain, as outlined in, e.g., Barriopedro et al. (2023) and Domeisen et al. (2023). However, simulations are commonly evaluated using averaged quantities such as HW frequencies or circulation anomalies averaged over one or all events. Changes in variability on short, HW intrinsic timescales are rarely taken into account. Although a link exists between the mean state and internal variability, their sensitivities to model deficiencies can be different, potentially indicating the importance of different error sources. Our investigation highlights day-to-day variability as a sensitive parameter in this respect. The substantial inconsistency in internal variability among the models points to shortcomings that are less obvious in the average quantities and thus further limits confidence in, for example, future projections of EHWs. The evaluation of models with regard to EHWs, and possibly also other extreme events at subseasonal scales, should therefore consider intrinsic variability as a component of the metrics.

*Author contributions.* DJ carried out the wave decomposition of CMIP5 models and reanalyses except ERA5. All other authors contributed to the study conception and design. IS performed the analysis and wrote a first draft of the manuscript. All authors participated in data interpretation and revised the manuscript. All authors read and approved the final manuscript.

*Competing interests.* The authors declare that the research was conducted in the absence of any commercial or financial relationships that could be construed as a potential conflict of interest.

*Acknowledgements.* We gratefully acknowledge Richard Blender for his suggestions and helpful discussions during the study, as well as constructive comments from Dr. Paolo De Luca and an anonymous reviewer. This work was funded by the Deutsche Forschungsgemeinschaft (DFG, German Research Foundation) under Germany's Excellence Strategy – EXC 2037 'CLICCS - Climate, Climatic Change, and Society' (CLICCS, A6) – Project Number: 390683824, contribution to the Center for Earth System Research and Sustainability (CEN) of Universität Hamburg.

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
