# Peer review of "Mean state and day-to-day variability of tropospheric circulation in planetary-scale barotropic Rossby waves during Eurasian heat extremes in CMIP5 models"

_EGUsphere, 2025_

## Referee Comment (RC1)

**Review Strigunova et al. WCD by Paolo De Luca**

The study by Strigunova et al. addresses the linkages between Eurasian heatwaves and Rossby waves. They used a small sample of CMIP5 models belonging to historical, AMIP and RCP4.5 experiments, along with ERA5 reanalysis. They show agreement between the models and ERA5 in reproducing surface temperature over Eurasia, EHWs and anomalies of Z500 during EHWs. However, there is little agreement between models and ERA5 when it comes to day-to-day variability of Rossby waves.

I personally find the paper very interesting and also timely, however there are several methodological, structural and written aspects of it that require the foremost attention of the authors, before the paper can be considered acceptable for publication. Please see below my comments.

Major comments

I think the abstract lacks a concluding sentence. Or, what are the implications of your study? Whether they are purely scientific of impact related.

L44 (and elsewhere) you refer to different metrics used for computing heat extremes. I think you should at least mention some of them to inform the reader about some of their differences. You can look here https://climpact-sci.org/indices/ for a general overview. Also, in our paper we used some of them: https://agupubs.onlinelibrary.wiley.com/doi/full/10.1029/2022GL102493

L62-77 this should go in the Methods section and adjust accordingly L78-81.

L83-86 should be removed and the details of the reanalysis used incorporated into Section 2.1 which can be renamed: "Reanalysis and CMIP5 datasets".

Section 2.1 specify the horizontal resolutions of the reanalysis and models, and if they have been regridded. Also make clear which variables did you use for the analysis.

The paper lacks a clear Methods and Data section. For instance in Section 2.2 from L111 there is a description of the method along with a presentation of some of the results, same in Section 3.1. Could you please confirm that this is a suitable format for WCD?

In the estimation of the EHW metrics you used two different periods: 1980-2005 and 2070-2100. Then in Table 1 you compare the number of HW days and HW events between these periods. I don't think this makes sense, because a longer period will likely contain more heatwave days and events. I think you should compare two periods of the same length, e.g. 1970-2000 and 2070-2100, so that you also compare the end of the two centuries.

Following my previous comment, in Figures 2-3 the climatologies of the models are computed for 1980-2005, whereas the ones of ERA5 for 1980-2019. That's a 14-year difference. Also climatology by definition isn't it minimum 30 years? I wonder if results change when considering two periods of the same length.

The statistical significance of the composites in Figures 3 and 4d-f is not assessed. This should be addressed with for example a parametric (or non-parametric) test between Z500 during EHWs and Z500 during non-EHWs.

Section 3.3. Although you refer to Setal2022 is not clear from the text to what type of energy you are referring to. Please make it clear in the section, although this should be described in the Methods section.

In Section 4 (Summary and conclusions) the study is not put into the broader perspective of the current literature, except for one link to "(e.g. Schaller et al., 2018; Brunner et al., 2018; Jeong et al., 2022)". The authors should discuss more the implications of their results and research gap addressed with their study by linking it to other studies.

Minor comments

L8 EHWs

L11 please add two key references supporting the statement

L12 you can add https://agupubs.onlinelibrary.wiley.com/doi/full/10.1029/2022GL102493

L18-19 word "bias" repeated trice in less than a row. Suggest to amend to "due to regional inaccuracies" and "atmospheric misrepresentation of teleconnections"

L26 CMIP5's

L27 "anomalies" repeated twice

L33 NH? Please define acronym if you use it later on, otherwise just write "northern hemisphere"

L37 "in relation to comparably" not clear, please rephrase

L38 double repetition of "trends"

L42-43 "that climate models"

L51-52 I think you can remove the sentence. Looks like a repetition of L39-40.

L55 "It coincides" what? The reduction of the RW variance? Please clarify

In general, when you refer to the models you used throughout the text, mention them as "CMIP5" and not as "CMIP".

L83 "for identifying"

L107 remove first sentence. Also, did not you use also the method from Ma and Franzke (2021)? Please calrify

L110 "we extend the set in by" ?

Section 2.1 add citations CMIP5, AMIP and RCP4.5

L139 "do not directly"

L141 "AMIP"

L142 "the boxplots". Also refer to figure 1.

L158 "the latter are" ?

L160 "Setal2000" ?

L168 change "heat waves" to "HWs"

Figure 4, make colorbar larger as the other ones.

L200 "Setel2022" or "Setal2022" ?

L207 "for all days HIST and AMIP" not clear.

Figure 6 caption: "for ERA5"

L218 EHWs are not defined only by T2m, but by other criteria too (e.g. persistence?). You can say "are defined by using near-surface….."

L219 if you mention the future period in the sentence you should also mention the present-day period.

L221 "reanalysis"

L225 "4"

L225-226 "it should be highlighted"

L232 "blocking pattern"

L238 "lead to the following"

L241 "Rossby"

L242 "reanalysis"

---

## Referee Comment (RC2)

**Review of Mean state and day-to-day variability of tropospheric circulation in planetary-scale barotropic Rossby waves during Eurasian heat extremes in CMIP models by Strigunova et al., egusphere-2025-892**

This study investigates how well a subset of CMIP5 climate models represent tropospheric barotropic Rossby waves during Eurasian heatwaves (EHWs), focusing on mean circulation patterns and day-to-day variability compared to ERA5 reanalysis. The authors apply a sophisticated normal-mode decomposition method to isolate planetary-scale waves and analyze differences between historical, AMIP, and RCP4.5 future simulations. The main finding is that while models capture mean circulation anomalies reasonably well, they fail to reproduce the observed changes in day-to-day variability (particularly skewness) during EHWs. Future projections show substantial uncertainty. Overall, the study addresses an important question and the approach is scientifically valuable, but there are several issues that require major attention before the manuscript is suitable for publication.

**Major comments:**

1) The manuscript includes qualitative comparisons between models and ERA5 in terms of 500 hPa geopotential height anomalies and wind composites during EHWs (Figs. 2-4), but does not include any quantitative metrics such as spatial correlation coefficients or RMS errors. As a result, the performance statements (e.g., AMIP runs "outperform" HIST runs) are not objectively substantiated.

2) The paper uses a subset of four CMIP5 models based on data availability for normal-mode decomposition. While this constraint is understandable, the manuscript does not assess whether these models are representative of broader CMIP5 behavior in terms of heatwave characteristics or large-scale circulation. For example, are these models typical or atypical in terms of blocking frequency or Z500 biases? Do they represent the CMIP5 ensemble well in Eurasian T2m skewness or Z500 amplitude? This is especially important because later conclusions (e.g., lack of skewness, success of AMIP) may depend heavily on model-specific features.

3) The paper finds that CMIP5 models do not capture the observed changes in PDF skewness of Rossby wave energy during EHWs, yet offers no discussion of possible physical or dynamical reasons for this failure. Skewness changes indicate nonlinear or intermittent behavior in the planetary wave field during EHWs, likely related to wave-mean flow interactions or blocking onset/stability. Not addressing the dynamical mechanisms weakens the interpretability of the results and their value for model improvement. I suggest the authors to discuss potential reasons why CMIP5 models fail

(e.g., too diffusive numerics, poor blocking representation, unresolved sub-monthly feedbacks) and to frame this failure as a window into dynamical model limitations, not just statistical mismatch.

**Other comments:**

- Title: I think it is better to clarify "CMIP models" to "CMIP5 models" since only CMIP5 subset is used.

- Line 4: "do not suggest an increase in EWHs" should be EHWs

- Final sentence of abstract: Lacks a conclusion or implication. I would add a final sentence summarizing why this matters (e.g., "This highlights key uncertainties in modelled dynamical variability under warming scenarios.").

- Line 33: Define "NH" (write "Northern Hemisphere") unless you will use it repeatedly later.

- Line 68-69: "do not directly affect" (not "not directly affect").

- Section 2: I suggest the authors to use a clear Data and Methodology section. More details of JRA-55, ERA-Interim, MERRA and ERA5 (e.g., variables) should be given under Reanalysis data subsection.

- The study uses inconsistent time periods: 1980-2005 for historical CMIP5, 2070-2100 for RCP4.5, and 1980-2019 for ERA5. These discrepancies affect the comparability of heatwave metrics and climatologies, especially when using absolute event counts. The manuscript should justify this choice clearly and consider using consistent or normalized periods for fair comparison.

- Line 90-91: I would consider more neutral phrasing: "Our subset reflects the models for which 3D data on model levels were available, without further selection criteria. "

- Methodology: Were reanalysis and model outputs regridded to a common grid before comparison? If so, please say so explicitly.

- Lines 104-105: Better to rephrase as "EHW events are identified by three or more consecutive days of positive anomalies…"

- Line 110: Rephrase as "we extended the set by adding the average duration."

- Line 112: The use of "boxes" to refer to boxplots is slightly informal. Rephrase as "The interquartile ranges shown in the boxplots... "

- AMIP should be spelled out at first use: "Atmosphere-only model simulations (AMIP)... "

- Lines 161-163: Consider rephrasing as "Troposphere-barotropic modes were identified by selecting vertical modes without zero-crossings within the troposphere, following the criteria used in Setal2022. "

- Line 180: Insert comma after "In contrast to CNRM-CM5"

- Line 207: Rephrase for clarity: "for all days in HIST and AMIP runs, respectively. "

- Line 238: Better to rephrase as "The results provide the following answers to the questions posed"

- Line 241: "Rssoby" to "Rossby"

- The issue of model biases impacting Rossby wave dynamics needs stronger discussion. For example, what does this imply for future projection reliability? Currently, it ends too cautiously. I would add a paragraph discussing what improvements would be needed in CMIP models to better capture day-to-day dynamics.

---

## Author Response (AR1)

Paper: wcd-2025-892, entitled "Mean state and day-to-day variability of tropospheric circulation in planetary-scale barotropic Rossby waves during Eurasian heat extremes in CMIP5 models", By Iana Strigunova, Frank Lunkeit, Nedjeljka Žagar, Damjan Jelić

**Response to the comments by Paolo De Luca**

Dear Dr. De Luca,

Thank you very much for your positive evaluation of our manuscript and your constructive comments and suggestions. Below please find our responses, presented in blue font following your comments in black font.

In addition, we have enclosed a draft of the revised manuscript, which largely incorporates the changes addressing your and other Reviewer's comments, as detailed in the point-to-point responses. An assessment of the statistical significance of the composites shown in Figures 3 and 4 (in response to the corresponding comment) is not yet included, as this requires further data processing. The corresponding changes will be incorporated in the final revised version, together with the improvement of the colorbar of Figure 4 (in response to the respective minor comment).

Your sincerely,

Iana Strigunova, Frank Lunkeit, Nedjeljka Žagar, Damjan Jelić

**Major comments**

I think the abstract lacks a concluding sentence. Or, what are the implications of your study? Whether they are purely scientific of impact related.

Response: We modified the end of the abstract, which now reads: "The associated Rossby wave circulation is considerably uncertain, with a particular lack of consistent representation of day-to-day variability. This further limits confidence in future projections of changes in EHWs. Our results suggest that intrinsic variability should be an additional component of the metrics evaluating the simulation of EHWs and their related circulation."

L44 (and elsewhere) you refer to different metrics used for computing heat extremes. I think you should at least mention some of them to inform the reader about some of their differences. You can look here <a href="https://climpact-sci.org/indices/">https://climpact-sci.org/indices/</a> for a general overview. Also, in our paper we used some of them: <a href="https://agupubs.onlinelibrary.wiley.com/doi/full/10.1029/2022GL102493">https://agupubs.onlinelibrary.wiley.com/doi/full/10.1029/2022GL102493</a>

Response: We expanded the respective paragraph on heat extreme metrics and included suggested paper. Now the paragraph reads as: "Existing uncertainties may be partially explained by the large number of metrics used to identify surface heat extremes, reflecting the complex interactions of underlying physical mechanisms (e.g. Horton et al., 2016; De Luca and Donat, 2023) and the varied needs of different scientific communities (Naomi et al., 2024). These diverse HW definitions often combine temperature with other meteorological variables like relative humidity: for instance, the use of wet-bulb temperature to compute the US Weather Service Heat Index from Buzan et al., (2015). Metrics vary based on absolute or relative thresholds and whether characteristics like duration, intensity, frequency, and spatial extent are considered (such as the Heat Wave Intensity Duration Frequency Curve from Mazdiyasni et al. (2019) or the Heat Wave Magnitude Index from Russo et al. (2014)). The specific goal of a study also influences the metric; for example, cumulative heat for health impacts (Perkins-Kirkpatrick and Lewis, 2020) or the timing of the HW season for ecosystem impacts (Sippel et al., 2016). To unify them in one framework, sets of indices are proposed by the Expert Team on Climate Change Detection (https://www.wcrp-climate.org/etccdi) and the Expert Team on Climate Information for Decisionmaking (https://climpact-sci.org/indices/). Furthermore, to assess future changes, some of these metrics are typically based on parameters estimated from present-day conditions (e.g. Sillmann et al., 2013b)."

L62-77 this should go in the Methods section and adjust accordingly L78-81.

Response: We expanded Data and Method section on the details and adjusted related parts of the text, as suggested by both Referees (see also below). We also kept most of this part (old L62-77), because, in our opinion, it provides a short introductory overview of the applied methodology.

L83-86 should be removed and the details of the reanalysis used incorporated into Section 2.1 which can be renamed: "Reanalysis and CMIP5 datasets".

Response: We followed Reviewer's suggestion and moved this part into the revised Data and Method section (Section 2, Subsection 2.1 Data).

Section 2.1 specify the horizontal resolutions of the reanalysis and models, and if they have been regridded. Also make clear which variables did you use for the analysis.

Response: This information is included in the revised Data and Method section.

The paper lacks a clear Methods and Data section. For instance in Section 2.2 from L111 there is a description of the method along with a presentation of some of the results, same in Section 3.1. Could you please confirm that this is a suitable format for WCD?

Response: The revised version includes Data and Methods and we restructured the text accordingly.

In the estimation of the EHW metrics you used two different periods: 1980-2005 and 2070-2100. Then in Table 1 you compare the number of HW days and HW events between these periods. I don't think this makes sense, because a longer period will likely contain more heatwave days and events. I think you should compare two periods of the same length, e.g. 1970-2000 and 2070-2100, so that you also compare the end of the two centuries.

Following my previous comment, in Figures 2-3 the climatologies of the models are computed for 1980-2005, whereas the ones of ERA5 for 1980-2019. That's a 14-year difference. Also climatology by definition isn't it minimum 30 years? I wonder if results change when considering two periods of the same length.

Response: We accept Reviewer's criticisms on this point. To account for the rareness of EHWs, we originally used the maximum lengths of the available reanalysis data during satellite era (i.e. 1980 onward), and similar for the AMIP and the HIST simulation (26 years), and 31 years (2071 to 2100) for RCP4.5. The latter two dataset were given from the availability of 3D scale-decomposed CMIP5 data. To account for the different sample sizes, in the revised version we normalised the total numbers of EHW events and EHW days to a 10-year period. In addition, we repeated the diagnostics (EHWs and 500 hPa circulation) for a 26-year period for all datasets (1980 to 2005 and 2075 to 2100, resp.). It turns out that the differences are small for the 500 hPa climatologies (see below) and the RCP4.5 simulations. However, the difference is significant for

ERA5 because of the extreme 2010 Russian heat wave. A comparable event is not present in the CMIP5 nor in the ERA5 datasets for the 26-year periods. This is discussed in the revised Data and Method section and in the respective results sections. We note that the 2010 Russian heat wave also affects the ERA5 EHW composites, but in a moderate way. This will be discussed in the context of the significance of the composites in the final revised version.

Figure 1 below shows the respective difference for the climatological ERA5 fields.

Figure 1. Climatological geopotential height (in gpm) ERA5 for two periods: (a) 1980-2005, (b) 1980-2019, (c) the difference between them. Please note the different scale (colorbar) for (c).

The statistical significance of the composites in Figures 3 and 4d-f is not assessed. This should be addressed with for example a parametric (or non-parametric) test between Z500 during EHWs and Z500 during non-EHWs.

Response: A measure of statistical significance will be added and discussed in the revised version.

Section 3.3. Although you refer to Setal2022 is not clear from the text to what type of energy you are referring to. Please make it clear in the section, although this should be described in the Methods section.

Response: We have added the following paragraph in the revised version: "In addition, the square of the absolute value of the complex expansion coefficient represents the total mechanical energy

of the particular mode, where the mechanical energy is the sum of kinetic and available potential energy (Zagar et al., 2015). Thanks to the 3D orthogonality of normal modes, the energies of the individual modes are additive. Energy anomalies are calculated relative to the climatology, which is defined for each calendar day of the extended boreal summer (MJJAS), and normalised by the climatological standard deviation (i.e. by variability). We use the time series of the mechanical energy to assess the day-to-day variability of troposphere-barotropic Rossby wave circulation associated with the EHWs in terms of the probability density function (PDF), in particular the skewness (Section 4.2)."

In Section 4 (Summary and conclusions) the study is not put into the broader perspective of the current literature, except for one link to "(e.g. Schaller et al., 2018; Brunner et al., 2018; Jeong et al., 2022)". The authors should discuss more the implications of their results and research gap addressed with their study by linking it to other studies.

Response: We have complemented summary and conclusion section by a discussion on possible sources of EHW biases as identified in previous studies, their relationship with day-to-day variability, and related implications for the model evaluation and for the reliability of future projections. Now, it reads as follows.

[revised manuscript text omitted]

**Minor comments**

L8 EHWs

The typo is corrected.

L11 please add two key references supporting the statement

The key references are added.

L12 you can add <a href="https://agupubs.onlinelibrary.wiley.com/doi/full/10.1029/2022GL102493">https://agupubs.onlinelibrary.wiley.com/doi/full/10.1029/2022GL102493</a>

The reference is added.

L18-19 word "bias" repeated twice in less than a row. Suggest to amend to "due to regional inaccuracies" and "atmospheric misrepresentation of teleconnections"

We followed the suggestion of changing to "due to regional inaccuracies", but we retained "atmospheric bias teleconnections" as this terminology follows Zhao et al. (2024) and denotes remote effects of local biases rather than an "atmospheric misrepresentation of teleconnections".

L26 CMIP5's

Corrected as suggested.

L27 "anomalies" repeated twice

Corrected as suggested.

L33 NH? Please define acronym if you use it later on, otherwise just write "northern hemisphere" We expanded the abbreviation.

L37 "in relation to comparably" not clear, please rephrase

We changed to "in relation to trends in the global mean surface temperature".

L38 double repetition of "trends"

Corrected as suggested.

L42-43 "that climate models"

Corrected as suggested.

L51-52 I think you can remove the sentence. Looks like a repetition of L39-40.

The phrase is removed in the revised text.

L55 "It coincides" what? The reduction of the RW variance? Please clarify

We rephrased it as "The reduction of the variance coincides".

In general, when you refer to the models you used throughout the text, mention them as "CMIP5" and not as "CMIP".

Corrected as suggested.

L83 "for identifying"

Corrected as suggested.

L107 remove first sentence. Also, did not you use also the method from Ma and Franzke (2021)? Please calrify

We use only one method which is similar to Ma and Franzke (2021) and Perkins-Kirkpatrick and Gibson (2017). This is clarified in the revised version in the new data and method section.

L110 "we extend the set in by"?

We corrected it as "we extended the set by adding the maximum temperature and the difference between the maximum temperature and the mean temperature".

Section 2.1 add citations CMIP5, AMIP and RCP4.5

In the revised version, we refer to Taylor et al. (2012).

L139 "do not directly"

Corrected as suggested.

L141 "AMIP"

Corrected as suggested.

L142 "the boxplots". Also refer to figure 1.

We corrected it as "the IQRs" and added a reference to figure 1.

L158 "the latter are"?

We changed it to "Index m".

L160 "Setal2000"?

Corrected as suggested.

L168 change "heat waves" to "HWs"

We changed it to "EHWs".

Figure 4, make colorbar larger as the other ones.

Colorbar size will be adjusted in the revised version. of the figure, along with the implemented statistical significance.

L200 "Setel2022" or "Setal2022"?

It should be "Setal2022". We corrected it in the text.

L207 "for all days HIST and AMIP" not clear.

We changed it to "all days in the HIST and AMIP runs, respectively".

Figure 6 caption: "for ERA5"

Corrected as suggested.

L218 EHWs are not defined only by T2m, but by other criteria too (e.g. persistence?). You can say "are defined by using near-surface...."

Corrected as suggested.

L219 if you mention the future period in the sentence you should also mention the present-day period.

Corrected as suggested.

L221 "reanalysis"

Corrected as suggested.

L225 "4"

Corrected as suggested.

L225-226 "it should be highlighted"

Corrected as suggested.

L232 "blocking pattern"

Corrected as suggested.

L238 "lead to the following"

We corrected it as "provide the following" according to another reviewer's suggestion.

L241 "Rossby"

Corrected as suggested.

L242 "reanalysis"

Corrected as suggested.

Paper: wcd-2025-892, entitled "Mean state and day-to-day variability of tropospheric circulation in planetary-scale barotropic Rossby waves during Eurasian heat extremes in CMIP5 models",

By Iana Strigunova, Frank Lunkeit, Nedjeljka Žagar, Damjan Jelić

**Response to the comments by Referee RC2**

Dear Referee,

Thank you very much for your positive evaluation of our manuscript and your constructive comments and suggestions. Below please find our responses, presented in blue font following your comments in black font.

In addition, we have enclosed a draft of the revised manuscript, which largely incorporates the reviewers' comments, as detailed in the point-to-point responses. An assessment of the statistical significance of the composites shown in Figures 3 and 4 (see the corresponding comment in Paolo De Luca's review) is not yet included, as this requires further data processing. The corresponding changes will be incorporated in the final revised version.

Your sincerely, Iana Strigunova, Frank Lunkeit, Nedjeljka Žagar, Damjan Jelić

**Major comments:**

1) The manuscript includes qualitative comparisons between models and ERA5 in terms of 500 hPa geopotential height anomalies and wind composites during EHWs (Figs. 2-4), but does not include any quantitative metrics such as spatial correlation coefficients or RMS errors. As a result, the performance statements (e.g., AMIP runs "outperform" HIST runs) are not objectively substantiated.

Response: Thank you for the comment. While our statements on the models' performance are objective, we agree that a quantitative metrics can further support our statements. In response to your comment, we included a quantitative comparison in a table (Table 3 in the revised paper), which provides Root Mean Square Errors (RMSEs)

and Anomaly Correlations (ACCs) for HIST and AMIP simulations (climatology and EHW anomalies) with respect to ERA5. The following paragraph has been added in the text: "For a more quantitative comparison, Table 3 provides the Root Mean Square Errors (RMSEs) and Anomaly Correlations (ACCs) for northern hemispheric 500-hPa climatologies of the CMIP5 HIST and AMIP simulations with respect to ERA5. These measures support our assessment by showing that AMIP simulations exhibit, with very few exceptions, higher values for ACC and lower values for RMSE compared to HIST."

"As for the climatology, the RMSEs and ACCs support our qualitative assessment, with better agreement with ERA5 for the AMIP simulations compared to HIST (Table 3, values in parentheses)."

**Table 3.** Anomaly Correlation (ACC) and Root Mean Squared Error (RMSE) for the CMIP5 HIST and AMIP simulations with respect to ERA5. RMSE and ACC are provided for the northern hemispheric 500-hPa climatologies, CLIM (shown in Fig. 2) and, in parentheses, for the respective difference between EHW composites and climatologies, DIFF (shown in Fig. S3 in the supplement).

| ACC/RMSE CLIM (DIFF)      | MPI-ESM-LR AMIP           | CNRM-CM5 HIST              | CNRM-CM5 AMIP               |
|---------------------------|---------------------------|----------------------------|-----------------------------|
| h'@500hpa                 | 0.93 (0.79) / 6.2 (8.13)  | 0.89 (0.12) / 8.98 (13.25) | 0.91 (0.62) / 7.18 (11.35)  |
| u'@500hpa                 | 0.9 (0.72) / 0.79 (0.82)  | 0.87 (0.36) / 0.99 (0.97)  | 0.83 (0.41) / 1.12 (1.27)   |
| v'@500hpa                 | 0.93 (0.79) / 0.29 (0.44) | 0.93 (0.11) / 0.4 (0.78)   | 0.94 (0.72) / 0.31 (0.55)   |
| GFDL-CM3 HIST             | GFDL-CM3 AMIP             | MIROC5 HIST                | MIROC5 AMIP                 |
| 0.89 (0.45) / 9.2 (11.38) | 0.91 (0.71) / 7.6 (10.4)  | 0.82 (0.24) / 14.1 (13.66) | 0.82 (0.68) / 13.94 (10.05) |
| 0.85 (0.27) / 1.3 (1.08)  | 0.92 (0.58) / 0.96 (1.12) | 0.83 (0.07) / 1.38 (1.3)   | 0.78 (0.36) / 1.53 (1.19)   |
| 0.9 (0.44) / 0.43 (0.64)  | 0.92 (0.74) / 0.33 (0.55) | 0.9 (0.5) / 0.78 (0.65)    | 0.92 (0.76) / 0.77 (0.48)   |

The paper uses a subset of four CMIP5 models based on data availability for normal-mode decomposition. While this constraint is understandable, the manuscript does not assess whether these models are representative of broader CMIP5 behavior in terms of heatwave characteristics or large-scale circulation. For example, are these models typical or atypical in terms of blocking frequency or Z500 biases? Do they represent the CMIP5 ensemble well in Eurasian T2m skewness or Z500 amplitude? This is especially important because later conclusions (e.g., lack of skewness, success of AMIP) may depend heavily on model-specific features.

Response: We thank Reviewer for this question. It is addressed by expanding the respective paragraph on our model subset (at the beginning of section 2.1) which now reads: "We use a subset of CMIP5 models that had outputs available on model levels to apply wave decomposition on terrain-following levels (see Section 2.3 below). No further selection criteria are applied. Our model subset consists of the CNRM-CM5 (Voldoire et al., 2013), the GFDL-CM3 (Donner et al., 2011), the MIROC5 (Watanabe et al., 2010) and the MPI-ESM-LR (Giorgetta et al., 2013). Although our selection is based only on the

availability of the data, we note that the four models are among the six models identified by Basharin et al. (2016) as climate models that best reproduce the historical behaviour of surface air temperature over greater Europe, selected from the CMIP5 project using a performance-based selection method.

Given the relatively small number, our model subset reasonably represents the spectrum of the CMIP5 simulations with regard to EHWs and atmospheric blockings. Concerning the EHWs, Hirsch et al. (2021) provide a thorough comparison of individual CMIP5 and CMIP6 models in their supporting information. Our four models appear to lie well within the range spanned by all CMIP5 models with respect to the bias skill scores for HW frequency, length of the longest HW, average HW intensity and cumulative heat. The same appears to be true for the representation of Northern Hemisphere blocking events. A comparison of the blocking frequencies of individual CMIP5 models including our four models is presented in the Supporting Information of Dunn-Sigouin and Son (2013), together with a comparison of 500-hPa zonal wind and variability."

3) The paper finds that CMIP5 models do not capture the observed changes in PDF skewness of Rossby wave energy during EHWs, yet offers no discussion of possible physical or dynamical reasons for this failure. Skewness changes indicate nonlinear or intermittent behavior in the planetary wave field during EHWs, likely related to wave-mean flow interactions or blocking onset/stability. Not addressing the dynamical mechanisms weakens the interpretability of the results and their value for model improvement. I suggest that the authors discuss potential reasons why CMIP5 models fail (e.g., too diffusive numerics, poor blocking representation, unresolved sub-monthly feedbacks) and frame this failure as a window into dynamical model limitations, not just statistical mismatch.

Response: As our study only identifies the potential bias in Rossby wave dynamics by statistical means, it does not provide physical causes or exact improvements needed to address shortcomings in the models that may also be different for each model. While this limits the far reach of our results, it complements earlier studies by highlighting importance of processes such as wave-mean flow interactions. At the end of the summary and conclusion section, we have added a discussion on possible sources of EHW biases as identified in previous studies, their relationship with day-to-day variability, and related implications for the model evaluation and for the reliability of future projections. Now, it reads as follows.

[revised manuscript text omitted]

**Other comments:**

- Title: I think it is better to clarify "CMIP models" to "CMIP5 models" since only CMIP5 subset is used.
  - We adjusted the title accordingly.
- Line 4: "do not suggest an increase in EWHs" should be EHWs. The typo is corrected.
- Final sentence of abstract: Lacks a conclusion or implication. I would add a final sentence summarizing why this matters (e.g., "This highlights key uncertainties in modelled dynamical variability under warming scenarios.").
  We modified the end of the abstract, which now reads: "The associated Rossby wave circulation is considerably uncertain, with a particular lack of consistent representation of day-to-day variability. This further limits confidence in future projections of changes in EHWs. Our results suggest that intrinsic variability should be an additional component of the metrics evaluating the simulation of EHWs and their related circulation."
- Line 33: Define "NH" (write "Northern Hemisphere") unless you will use it repeatedly later.
   We expanded the abbreviation.
- Line 68-69: "do not directly affect" (not "not directly affect").
   We found the expression in Line 139 and corrected it.

- Section 2: I suggest the authors to use a clear Data and Methodology section. More details of JRA-55, ERA-Interim, MERRA and ERA5 (e.g., variables) should be given under Reanalysis data subsection.
  - We added a Data and Method section and adjusted related parts of the text, as suggested by both referees. In the new section we explain the use of the other three reanalyses.
- The study uses inconsistent time periods: 1980-2005 for historical CMIP5, 2070-2100 for RCP4.5, and 1980-2019 for ERA5. These discrepancies affect the comparability of heatwave metrics and climatologies, especially when using absolute event counts. The manuscript should justify this choice clearly and consider using consistent or normalized periods for fair comparison.
  - To account for the rareness of EHWs, we used the maximum lengths of the available data sets for the reanalyses, the AMIP and the HIST simulation (26 years), and 31 years (2071 to 2100) for RCP4.5. The latter two dataset were given from the availability of 3D scale-decomposed CMIP5 data. To account for the different sample sizes, in the revised version we normalised the total numbers of EHW events and EHW days to a 10-year period. In addition, we repeated the diagnostics (EHWs and 500 hPa circulation) for a 26-year period for all data sets (1980 to 2005 and 2075 to 2100, resp.). It turns out that the differences are small for the 500 hPa climatologies (see below) and the RCP4.5 simulations. However, the difference is significant for ERA5 because of the extreme 2010 Russian heat wave. A comparable event is not present in the CMIP5 nor in the ERA5 datasets for the 26-year periods. This is described in the new Data and Method section and the respective results sections. We note that the 2010 Russian heat wave also affects the ERA5 EHW composites, but in a moderate way. This will be discussed in the context of the significance of the composites in the final revised version.

(a) ERAS MIJAS (1980-2019)
(b) ERAS MIJAS (1980-2005)
(c) Difference (b) - (a)

Figure 1 below shows the respective difference for the climatological ERA5 fields.

Figure 1. Climatological geopotential height (in gpm) ERA5 for two periods: (a) 1980-2005, (b) 1980-2019, (c) the difference between them. Please note the different scale (colorbar) for (c).

Line 90-91: I would consider more neutral phrasing: "Our subset reflects the models for which 3D data on model levels were available, without further selection criteria."

We adjusted the respective sentence in the course of answering the second major comment (see above).

 Methodology: Were reanalysis and model outputs regridded to a common grid before comparison? If so, please say so explicitly.

We used the Eurasian area-averaged near-surface temperatures on the individual input grids to assess the EHWs. For the 500hPa circulation, we regridded all data sets to the 256 by 128 grid of the reanalyses. This is now clarified in the Data and Method section: "Following Setal2022, we identify EHWs using the mean daily T2m averaged over Eurasia. Averaging is done using data on the individual input grids." and "For this part, all data sets are regridded to the 256 by 128 grid of the reanalyses."

 Lines 104-105: Better to rephrase as "EHW events are identified by three or more consecutive days of positive anomalies..."
 We rephrased it in the text.

- Line 110: Rephrase as "we extended the set by adding the average duration." We rephrased it in the text.
- Line 112: The use of "boxes" to refer to boxplots is slightly informal. Rephrase as "The interquartile ranges shown in the boxplots... "

  We rephrased it in the text.
- AMIP should be spelled out at first use: "Atmosphere-only model simulations (AMIP)... "

We expanded the abbreviation.

• Lines 161-163: Consider rephrasing as "Troposphere-barotropic modes were identified by selecting vertical modes without zero-crossings within the troposphere, following the criteria used in Setal2022."

We rephrased it in the text.

- Line 180: Insert comma after "In contrast to CNRM-CM5" The comma is added.
- Line 207: Rephrase for clarity: "for all days in HIST and AMIP runs, respectively. " We rephrased it in the text.
- Line 238: Better to rephrase as "The results provide the following answers to the questions posed"

We rephrased it in the text.

- Line 241: "Rssoby" to "Rossby" The typo is corrected.
- The issue of model biases impacting Rossby wave dynamics needs stronger discussion. For example, what does this imply for future projection reliability? Currently, it ends too cautiously. I would add a paragraph discussing what improvements would be needed in CMIP models to better capture day-to-day dynamics.

Please see our response to the third major comment.

---

## Author Response (AR2)

Paper: wcd-2025-892, entitled "Mean state and day-to-day variability of tropospheric circulation in planetary-scale barotropic Rossby waves during Eurasian heat extremes in CMIP5 models",

By Iana Strigunova, Frank Lunkeit, Nedjeljka Žagar, Damjan Jelić

**Response to the comments by Referee RC2**

Dear Referee,

Thank you very much for your positive evaluation of our manuscript and your constructive comments and suggestions. Below please find our responses, presented in blue font following your comments in black font.

In addition, we have enclosed the revised manuscript, which largely incorporates the reviewers' comments, as point-to-point responses. Please note that we have added an assessment of the statistical significance of the composites shown in Figures 3 and 4 (as well as Fig. S3 and S5) in this version.

Your sincerely,
Iana Strigunova, Frank Lunkeit, Nedjeljka Žagar, Damjan Jelić

Minor comments:

Lines 8-9: Better to rephrase to clarify that it is the model representation that is uncertain: "There is considerable uncertainty in how climate models represent the associated Rossby wave circulation…"
Response: We rephrased as suggested.

Line 9: "This further limits confidence in future projections…": Unclear whether "this" refers to circulation uncertainty, day-to-day variability, or model disagreement.
We rephrased it as "There is considerable uncertainty in how climate models represent the associated Rossby wave circulation".

Lines 50-51: Consider integrating the citation directly into the sentence for smoother flow: "...wet-bulb temperature, as used by Buzan et al. (2015) to compute the US Weather Service Heat Index."
The citation is integrated as suggested.

Line 123: "35a period" should be "35-year period"

Corrected as suggested.

Line 125: "We use the entire periods (40 and 35 years, resp.) to consider the largest possible datasets, given the rarity of EHWs.": could be simplified to "We use the full periods to maximize the sample size, due to the rarity of EHWs."

Corrected as suggested.

Section titles are inconsistently capitalized. For example, "2 Data and Methods" uses title case, while others like "3 The statistics of Eurasian surface heat waves" do not.

We corrected it, so now only the first word is capitalised.

Paper: wcd-2025-892, entitled "Mean state and day-to-day variability of tropospheric circulation in planetary-scale barotropic Rossby waves during Eurasian heat extremes in CMIP5 models",

By Iana Strigunova, Frank Lunkeit, Nedjeljka Žagar, Damjan Jelić

**Response to the comments by Roberto Rondanelli**

Dear Roberto Rondanelli,

Thank you very much for your positive evaluation of our manuscript and your constructive comments and suggestions. Below please find our responses, presented in blue font following your comments in black font.

In addition, we have enclosed the revised manuscript, which largely incorporates the reviewers' comments, as point-to-point responses. Please note that we have added an assessment of the statistical significance of the composites shown in Figures 3 and 4 (as well as Fig. S3 and S5) in this version.

Your sincerely,
Iana Strigunova, Frank Lunkeit, Nedjeljka Žagar, Damjan Jelić

Minor comments:

line 21: causing biases in atmospheric teleconnections
We prefer "atmospheric bias teleconnections" as this terminology follows the reference Zhao et al. (2024) and denotes remote effects of local biases rather than a "bias in atmospheric teleconnections".

line 22: and other anthropogenic
line 39: and in a regional
Corrected as suggested.

I am not entirely happy with the readability of Figure 1 in particular with the way in which the orange bars can be seen in the HIST/RCP4.5 comparison. I ask the authors to decide if it's better to move the RCP4.5 bars to a single row.
The new figure with orange bars as a single row is in the revised version.

line 224 over western North America
Corrected as suggested.

Also I apologize that this suggestion should have come earlier in the process, but I think Figures 2 to 4 would benefit from slightly thinner coastal boundaries and auxiliary lat lon lines, and also perhaps slightly thinner vectors, so anomalies are better observed.
New figures with incorporated suggestions are in the revised version.

Finally, I wish authors would develop an explanation (even if it's speculative) on why AMIP simulations perform better than HIST ones in the summary and conclusions.
We added the following sentences in the conclusions:
"This is consistent with the outperformance of AMIP compared to HIST simulations found in our study. These results seem to indicate that the background state is more important for the EHWs than the interaction with the ocean, at least in the limit of the differences between the mean states simulated in AMIP and HIST. A thorough assessment of the role and the importance of the atmosphere-ocean coupling can thus only be achieved with comparably good climatologies in coupled and uncoupled simulations."